# OpenReview forum: "ETA: Evaluating Then Aligning Safety of Vision Language Models at Inference Time"
_ICLR.cc/2025/Conference — ICLR 2025 Poster_

### Official Review · Reviewer_JJRX · 2024-10-27

**Soundness:** 3
**Presentation:** 4
**Contribution:** 3
**Rating:** 6
**Confidence:** 5

**Summary:**

The paper proposes a two-phase approach called "Evaluating Then Aligning" (ETA), aiming at improving the safety and utility of Vision Language Models (VLMs) during inference. It begins by assessing the safety of both visual and textual inputs and outputs using multimodal evaluation. If unsafe behavior is detected, ETA applies a bi-level alignment strategy to ensure responses are safe and useful.  Overall, ETA offers a plug-and-play solution that requires no additional training, making it a practical approach for real-world applications of VLMs.

Key Contributions:

(1) The framework uses visual content analysis and a reward model to evaluate the safety of inputs and outputs, creating a more robust safety awareness across both modalities.

(2) The method features shallow alignment using an interference prefix to activate safety mechanisms, followed by deep alignment with sentence-level best-of-N searching to ensure helpfulness without compromising safety.

(3) Extensive experiments demonstrate that ETA significantly reduces unsafe responses across various VLMs and benchmarks, outperforming existing inference-based methods. It maintains the general capabilities of the models while enhancing safety and efficiency.

**Strengths:**

Originality:

The paper introduces a novel framework, ETA (Evaluating Then Aligning), which addresses a critical gap in the safety of Vision Language Models (VLMs) by focusing on inference-time alignment. Unlike existing approaches that primarily rely on extensive fine-tuning or are limited to specific types of inputs, ETA offers a fresh perspective by combining multimodal evaluation and bi-level alignment without requiring additional training. This plug-and-play nature makes it a highly original contribution, providing a more flexible and scalable solution for enhancing VLM safety.

Quality:

The technical quality of the paper is robust, with a well-structured methodology that is clearly articulated and backed by extensive experimental validation. The authors systematically evaluate ETA across multiple dimensions, including safety, helpfulness, and efficiency. The experiments are comprehensive, covering various adversarial benchmarks and showing consistent improvements over baseline methods.

Clarity:

The paper is clearly written. The framework’s methodology is detailed and easy to follow, aided by diagrams that illustrate key concepts, such as the two-phase evaluation and alignment process. Additionally, the use of concise and relevant examples helps in understanding how ETA addresses existing limitations.

Significance:

The significance of this work is substantial, as it addresses one of the key challenges in deploying VLMs in real-world applications—ensuring safety without compromising utility. By introducing a method that does not require fine-tuning, ETA can be easily integrated into existing systems, making it practical for widespread use. The ability to improve safety while maintaining model efficiency and general capabilities could encourage broader adoption of VLMs.

**Weaknesses:**

1. Limitations in Pre-Generation Evaluation Using CLIP

A notable weakness of the paper lies in the reliance on CLIP for pre-generation safety evaluation, specifically through the use of predefined prompts (e.g., checking for "physically harmful" content). While CLIP is a powerful model for general vision-language tasks, its training data may not always align well with the specific categories being evaluated, such as physical harm. This could lead to situations where CLIP fails to accurately assess certain types of harmful content because it was not explicitly trained on safety-critical datasets. To improve this, the authors could consider using adaptive prompts that are tailored to different scenarios, enabling more precise evaluations. By refining prompts based on the context of the input (e.g., using distinct prompts for detecting physical hazards, offensive imagery, or illegal content), the system could achieve more accurate safety checks across a wider range of situations.

2. Potential Instability in Post-Generation Evaluation Using Reward Models (RMs)

The use of reward models (RMs) for post-generation evaluation could introduce instability, as the ability of RMs to accurately judge safety and utility might vary depending on the context. The RM employed may not be inherently designed to handle safety-specific assessments, which raises concerns about the consistency and reliability of its evaluations. Furthermore, this approach can lack interpretability, as it does not always provide clear reasons for why a response was considered safe or unsafe. To address this, it would be beneficial to either fine-tune the RM specifically for safety-related tasks or incorporate explainability mechanisms that can offer insights into how safety decisions are being made, thus building more trust in the evaluation process. For instance, the model could be designed to generate detailed explanations for its safety assessments, which could then be evaluated themselves to ensure they align with safety standards.

**Questions:**

(1) How can the interpretability of the post-generation evaluation process be improved?

The reliance on RMs can sometimes lead to a lack of clear explanations for why a response is deemed safe or unsafe. Have you considered methods to enhance interpretability, such as generating explanations alongside safety scores, or using models that can provide reasoning for their evaluations?

(2) Did you observe any scenarios where the ETA framework failed or produced unexpected results?

Understanding the limitations of ETA would be useful for assessing its real-world applicability. Are there specific types of inputs or adversarial attacks where ETA was less effective? For example, were there instances where CLIP misinterpreted visual content, leading to unsafe outputs, or where the RM failed to accurately judge the safety of a response? Additionally, if such limitations were observed, what improvements or adjustments could be considered to address these cases effectively?

---

> ### Author Response · Authors · 2024-11-21
> **Response to Reviewer JJRX (1/1)**
>
> We sincerely thank the reviewer for your valuable comments and appreciate your recognition of the novelty, flexibility, and scalable of ETA as well as the superiority of our work. We believe the constructive feedback will improve the paper and increase its potential impact on the community.
>
> > - Weakness 1: Limitations in Pre-Generation Evaluation Using CLIP
>
> Thanks for the constructive suggestions. We found that the original CLIP model already performs strongly in our task even without fine-tuning. This observation aligns with previous works that highlight its strong generalization ability in safety-related downstream tasks [1]. Specifically, we have the following findings:
>
> (i) **CLIP score effectively distinguishes between harmful and safe images:** As shown in Fig. 3(b), we found that CLIP score effectively distinguishes between harmful and safe images.
>
> (ii) **Removing some categories in the prompt does not compromise CLIP’s performance:** Our safety-related prompts already include common safety categories such as "physical harm". However, even after removing some of these categories, the CLIP score still effectively distinguishes between harmful and safe images. We believe this is because, as long as the textual input is semantically harmful, regardless of the specific unsafe categories, the CLIP-encoded image associated with harmful content naturally receives a higher CLIP score compared to non-harmful images.
>
> (iii) **The goal is to identify whether the image is safe, rather than classify its unsafe categories:** Moreover, the purpose of our Pre-Generation Evaluator is simply to assess whether the input image is harmful or not, rather than categorize it into specific harmful categories. As demonstrated in our experiments, the CLIP score is effective in detecting unsafe images through our designed prompt.
>
> > -  Weakness 2 & Question 1: Potential Instability in Post-Generation Evaluation Using Reward Models (RMs)
>
> Thanks for the valuable suggestions. To address the instability of the reward model in safety evaluation, we have proposed a safety-specific input format, as shown in Appendix A.2. The reward model we used already shows strong capabilities in safety evaluation [2]. We found that this designed input format further enhances the Reward Model’s ability to distinguish between safe and unsafe responses. Specifically, the safety-specific input format significantly increased the KL divergence between the distributions of safe and unsafe responses, **from 0.07 to 0.21**.
>
> The focus of ETA is not on improving the interpretability of the reward model, but rather on utilizing it to distinguish between safe and harmful responses. ETA can also be combined with judge models capable of generating explanations, and we leave this exploration for future work.
>
> We are also looking forward to the release of more safety-specific reward models or VLM reward models in the future. As a flexible plug-and-play framework, ETA can easily incorporate more powerful reward models, leading to even better performance.
>
> > - Question 2: Did you observe any scenarios where the ETA framework failed or produced unexpected results?
>
> As shown in our experimental results, while ETA can serve as a plug-and-play method to significantly reduce the USR compared to baselines, there are still instances where it struggles. We hypothesize that this is mainly due to **the performance of VLM baselines**. As a plug-and-play method, the alignment phase still relies on the VLM baseline to generate responses. The baseline’s inherent capabilities, such as instruction-following ability and built-in safety measures, will impact the performance of ETA. As shown in the table below, we observed that when ETA is applied to **Llama3.2-11B-Vision-Instruct (released in Sep. 2024)** a more recent and stronger model, the reduction in USR is significantly greater than that of other baselines. This finding highlights the potential of ETA, as future models with enhanced capabilities are released, ETA can be seamlessly applied to safeguard new VLM baselines.
>
> | Method  | SPA-VL Harm | MM-SafetyBench | FigStep (Vanilla) | FigStep (Suffix) | Adv. Image+Text |
> | :- | :-: | :-: | :-: | :-: | :-: |
> | LLaVA-NeXT-8B | 23.02 | 30.18 | 49.40 | 63.40 | 62.50 |
> | + ETA | **11.32** | **10.48** | **20.60** | **19.60** | **17.50** |
> | LLaVA-OneVision-7B-Chat | 15.85 | 29.76 | 45.20 | 40.40 | 70.00 |
> | + ETA | **6.79** | **10.60** | **16.80** | **19.40** | **20.00** |
> | Llama3.2-11B-Vision-Instruct | 7.17 | 19.17 | 41.60 | 44.00 | 15.00 |
> | + ETA | **2.64** | **3.99** | **8.20** | **3.20** | **7.50** |
>
> ## Reference
>
> [1] Tsai, Wei Lun, Jacob J. Lin, et al. Generating construction safety observations via CLIP-based image-language embedding. European Conference on Computer Vision. Cham: Springer Nature Switzerland, 2022.
>
> [2] Haoxiang Wang, Wei Xiong, et al. Interpretable preferences via multiobjective reward modeling and mixture-of-experts. arXiv preprint arXiv:2406.12845, 2024

---

> > ### Comment · Reviewer_JJRX · 2024-11-22
> >
> > Thank you for your explanation.
> > What I want more is for VLM to refuse to answer malicious questions as much as possible, even if it means responding with a generic reply like "I am sorry," rather than being compromised. That said, ASR doesn’t seem as effective as non-inference-time methods, so I feel the room for improvement with such inference-time approaches is relatively limited.
> > As such, I will keep my rate at 6.

---

> > > ### Author Response · Authors · 2024-11-22
> > >
> > > Thanks a lot for your reply. We appreciate your insightful comments, which improve our work.
> > >
> > > In our experiments (Table 2 and Table 11), fine-tuning-based methods like VLGuard, while achieving lower ASR, suffer from over-conservativeness. This results in a **significant degradation of the model’s general ability**, particularly in its performance on comprehensive benchmarks. In contrast, ETA preserves the model’s general capabilities while offering enhanced safeguards for VLM. This improvement is achieved **without requiring extra data or additional training**, highlighting ETA’s advantage over fine-tuning-based methods.
> > >
> > > In fact, for users who prioritize low ASR over helpfulness, ETA can achieve it by adjusting the Multimodal Evaluator's threshold. However, this trade-off may not be practical, as excessive conservativeness can impact the model’s usability for normal tasks.
> > >
> > > Finally, ETA can complement fine-tuning-based models to further reduce ASR. For instance, Llama3.2-11B-Vision-Instruct, which employs **RLHF**, achieves low ASR but still struggles with certain challenging cases where harmful responses are generated. By integrating ETA, the USR can be further reduced. For example, on FigStep (Suffix), the USR significantly decreased from **44.00 to 3.20** with the application of ETA.

---

> > > ### Author Response · Authors · 2024-11-25
> > > **A kindly reminder**
> > >
> > > Dear Reviewer,
> > >
> > > We are looking forward to your professional suggestions and hope to further discuss and refine the contents of the work.
> > >
> > > We sincerely appreciate your time and effort in helping us improve our work.
> > >
> > > Sincerely,
> > >
> > > Authors

---

> > > > ### Comment · Reviewer_JJRX · 2024-11-26
> > > >
> > > > Thanks. I believe your overall framework works well, which is the reason why I gave the positive score.
> > > >
> > > > One reason for why I cannot give a higher score is this two components within your framework are not that novel, one is using CLIP to detect malicious visual cues, the other is using best-of-n strategy to guarantee the most appropriate output (led by the predefined prompt like "As an AI assistant"). Could you explain further on what's your differences between existing works?

---

> > > > > ### Author Response · Authors · 2024-11-27
> > > > >
> > > > > Thank you for recognizing our work. Your feedback has been instrumental in helping us improve it further. While ETA leverages existing methods like CLIP and Best-of-N, its key contribution lies in the novel and unique ways these techniques are used to address VLM safety.
> > > > >
> > > > > To the best of our knowledge, [1] is the only prior work using CLIP in VLMs, where CLIP score is computed between output text and input image to construct preference datasets for DPO **fine-tuning**, aimed at reducing VLM hallucination. In contrast, we use CLIP for VLM safety, which differs significantly from prior work, as summarized below:
> > > > >
> > > > > **(i): Using CLIP for multimodal evaluation:** Our evaluation phase leverages a **key insight**: the **continuous visual embeddings** in VLMs, which are prone to bypassing safety mechanisms, are essentially **derived from the pre-trained CLIP vision encoder**. This inspired us to **innovatively** use the CLIP score to evaluate whether an image contains harmful semantics, enabling visual safety awareness for VLMs.
> > > > >
> > > > > **(ii): Using CLIP for inference-time Best-of-N:** In the alignment phase, we use the CLIP score as part of the criterion for sentence-level Best-of-N searching. Relying solely on the Reward score (criterion in vanilla Best-of-N) can result in generic rejection responses that lack user-specific helpfulness, while relying solely on the CLIP score (criterion in [1]) risks generating harmful content closely relevant to the image. In Eq. 6, we combine the CLIP score and Reward score in a **fine-grained manner as criteria**, ensuring that while providing safe responses, more helpful suggestions are also offered, which is difficult to achieve in previous safety alignment work.
> > > > >
> > > > > We are not aware of the use of Best-of-N (BoN) in VLMs. Previous work has applied BoN to LLMs, typically using criteria like Reward score, and selecting from complete responses [2]. Our work is the **first** to use BoN in VLMs, with the following novelties:
> > > > >
> > > > > **(i): Sentence-level BoN for improved safety:** As our experiments in Fig. 10 show, even with shallow alignment, directly generating a complete response can lead to harmful content due to the occurrence of transition words like "However." This issue **cannot be avoided in vanilla response-level Best-of-N**. Sentence-level Best-of-N offers **finer-grained** selection, enabling us to identify better trajectories and promptly terminate candidates when harmful semantics emerge.
> > > > >
> > > > > **(ii): Combining CLIP and Reward scores as BoN selection criteria:** Since there is **no suitable open-sourced VLM Reward Model**, we **innovatively** use the CLIP score (utility guide) and the Reward score (safety guide) together as selection criteria of BoN, ensuring both safety and helpfulness.
> > > > >
> > > > > **(iii): Sentence-level BoN enables fine-grained selection criteria:** As described in lines 351-356 of the paper, we combine the CLIP score and Reward score in different ways at different sentence indices to achieve a more **fine-grained selection**. For example, during the generation of the first candidate sentence, we use only the Reward score to ensure the overall safety of the response. In subsequent sentence generations, we incorporate the CLIP score with a weight of 1/i to improve the helpfulness of the response.
> > > > >
> > > > > ## Reference
> > > > > [1] Zhou Y, Fan Z, Cheng D, et al. Calibrated self-rewarding vision language models. NeurIPS 2024.
> > > > >
> > > > > [2] Brown, Bradley, et al. Large language monkeys: Scaling inference compute with repeated sampling. arXiv 2024.

---

> > > > > ### Author Response · Authors · 2024-12-01
> > > > > **A kindly reminder**
> > > > >
> > > > > Dear Reviewer,
> > > > >
> > > > > Thank you again for your constructive feedback on our work.
> > > > >
> > > > > We have **summarized the differences** between our use of CLIP score and Best-of-N and the previous work, as well as **our innovations**. Given that the discussion phase is coming to an end, please let us know if you have any remaining questions or concerns.
> > > > >
> > > > > We sincerely appreciate your time and effort in helping us improve our work.
> > > > >
> > > > > Sincerely,
> > > > >
> > > > > Authors

---

### Official Review · Reviewer_tnBM · 2024-10-31

**Soundness:** 3
**Presentation:** 3
**Contribution:** 2
**Rating:** 5
**Confidence:** 4

**Summary:**

The paper proposed a new safeguard mechanism for the Vision Language Model (VLM) during inference phase. This simple and straightforward approach largely reduces the unsafe responses while ensuring the helpfulness corresponding to the given input instruction.

**Strengths:**

- The proposed safeguard mechanism induces helpfulness, which better promotes usefulness of VLMs in a safe manner, rather than simply refusing the given user instruction.
- The paper is well written to easily understand
- The defense performance is well-improved compared to the baseline defense methods.

**Weaknesses:**

- Novelty: although the simplicity of the proposed methods, the novelty seems significantly limited, as the proposed methods are based on the simple application of existing models such as CLIP and Reward Models.
- Clarification of the motivation: While the authors posed the major cause of the VLMs being jailbroken is the continuous nature of the visual embeddings, I think it is due to the visual embeddings not being safety-aligned with RLHF. Also, I'm not sure if the proposed methods correspondingly resolve the motivation behind the visual embedding's continuous nature.
- Potential side-effects of the proposed method:
   1) Pre-generation Evaluator: although the image contains the harmful image, the textual instruction might be normal and benign (e.g., a criminal scene image + "How to prevent such crimes?"). According to the proposed method, it may judge these benign inputs as unsafe, which possibly leaves as a side-effect.
    2) Sentence-level Deep Alignment: since the proposed deep alignment selects the optimal sentence under the consideration of the safeness of the given sentence and its correlation with the input image in a greedy manner, the resultant sequence of sentences might not be ensured to be logically connected. This potential side-effect might rather hinder the reliability of the VLMs.

**Questions:**

- Performance comparison with the safety-aware visual RLHF (if any) would be appreciated to see if the proposed method would rather be the optimal way of safeguard VLMs.

---

> ### Author Response · Authors · 2024-11-21
> **Response to Reviewer tnBM (1/3)**
>
> We sincerely thank the reviewer for your valuable comments and appreciate your recognition of the well-written and superior performance including the safety and helpfulness of our work. We believe your constructive feedback will improve the paper.
>
> > - Weakness 1: Novelty: although the simplicity of the proposed methods, the novelty seems significantly limited, as the proposed methods are based on the simple application of existing models such as CLIP and Reward Models.
>
> While ETA leverages existing models such as CLIP and reward models (RMs), its contribution lies in how these models are uniquely used to address VLM safety, which we will further elaborate on below. In fact, ensuring safety while preserving the original capabilities of VLMs is a very challenging problem, and we believe this is where ETA makes a significant contribution. We summarize the novelty of ETA as follows:
>
> (i) **New Perspective on Safety Issues in VLMs:** We validate that the primary factor contributing to safety problems in VLM systems is the continuous nature of visual token embeddings.
>
> (ii) **First to Introduce Additional Visual Safeguards:** The usage of CLIP in our work differs significantly from its application in prior studies. Previous works have primarily leveraged the CLIP score during post-processing stages, such as inference and evaluation, to assess the similarity between output text and input images. In contrast, our paper innovatively utilizes the CLIP model during the pre-processing stage. We demonstrate that the CLIP score can be effectively generalized to evaluate the **safety** of continuous visual embeddings, which can easily bypass the existing safety mechanisms in VLMs. By utilizing predefined prompts with unsafe semantics, we show that it is generalizable to classify input images for safety directly, providing a novel application of CLIP in ensuring the safety of visual inputs.
>
> (iii) **An innovative Strategy to Improve Safety Evaluation Capability of Reward Models:** We demonstrate that designing a safety-specific input format for general reward models can significantly enhance their ability to more effectively evaluate the safety of responses. This differs from RMs’ usage in prior studies.
>
> (iv) **Harmless and Helpful Responses, Rather Than Generic Refusals:** We found that existing methods often provide generic refusals such as “I cannot help you, …”. However, an ideal response should be both harmless and helpful, offering helpful suggestions to users. By leveraging the multimodal evaluator (CLIP+RM) for sentence-level Best-of-N searching, we can effectively identify safer and more helpful generation trajectories. This is also a novel usage of CLIP and RM.
>
> (v) **Superior Performance of ETA as a Simple Plug-and-Play Framework:** Extensive experiments demonstrate that our plug-and-play approach is applicable to common VLM baselines, significantly enhancing the model’s safety without affecting its general capabilities. Moreover, it improves both the safety and usefulness of the model’s responses in unsafe scenarios.

---

> ### Author Response · Authors · 2024-11-21
> **Response to Reviewer tnBM (2/3)**
>
> > - Weakness 2 & Questions 1: Clarification of the motivation: While the authors posed the major cause of the VLMs being jailbroken is the continuous nature of the visual embeddings, I think it is due to the visual embeddings not being safety-aligned with RLHF. Also, I'm not sure if the proposed methods correspondingly resolve the motivation behind the visual embedding's continuous nature. & Performance comparison with the safety-aware visual RLHF (if any) would be appreciated to see if the proposed method would rather be the optimal way of safeguard VLMs.
>
> Thanks for your valuable comment.
>
> (i) We believe these two explanations are not contradictory. As shown in Figure 2, we illustrate that continuous visual token embeddings bypass existing safety mechanisms, which were **aligned based on discrete text token embeddings**. This highlights the need to establish safety awareness specifically for these outlier continuous visual tokens. While applying RLHF to these visual embeddings could be a potential solution, **it is resource-intensive**. It requires a substantial number of annotators, the collection of difficult-to-obtain multimodal datasets, and significant computational resources for training. Consequently, **most open-source VLMs have not undergone safety-aware visual RLHF and lack safety awareness for continuous visual embeddings**. Therefore, we propose an alternative solution, ETA, which is an **inference-time plug-and-play** framework, demonstrating excellent performance across various baselines.
>
> (ii) Our solution correspondingly resolve the motivation behind the visual embedding's continuous nature. We note that these continuous visual embeddings in VLMs are essentially obtained by the pre-trained CLIP vision encoder. CLIP is trained to align vision and text embeddings by maximizing semantic similarity across modalities [1]. This inspired us to leverage CLIP score to evaluate the safety of continuous visual embeddings by measuring their similarity to pre-defined harmful text prompts embeddings. This approach requires only the additional use of the CLIP text encoder to effectively assess whether the input image is safe. Therefore, we **enhance VLMs’ multimodal safety awareness by utilizing the CLIP score to assess the safety of visual inputs**.
>
> (iii) **At the time of submission, no VLMs with safety-aware visual RLHF were known to exist**. Now, we have identified a recently released model Llama3.2-11B-Vision-Instruct (released in Sep. 2024), which incorporates safety-aware visual RLHF. Since Llama3.2-11B-Vision-Instruct does not have a corresponding model without RLHF, we choose another widely-used VLM LLaVA-1.5-13B and apply ETA on it to compare ETA and visual RLHF. The results, as shown in the table below, reveal that applying ETA leads to originally weaker safety performance of LLaVA-1.5-13B resulting in a USR that is generally lower than that of Llama3.2-11B-Vision-Instruct. In addition, we also **applied ETA to Llama3.2-11B-Vision-Instruct and found that applying ETA further enhances the safety** of the baseline aligned with RLHF. This demonstrates the promising application potential of ETA.
>
> | Method  | SPA-VL Harm | MM-SafetyBench | FigStep (Vanilla) | FigStep (Suffix) | Adv. Image+Text |
> | :------- | :----: | :----: | :----: | :----: | :----: |
> | LLaVA-1.5-13B | 40.75 | 41.01 | 61.60 | 66.40 | 100.00 |
> | Llama3.2-11B-Vision-Instruct | 7.17 | 19.17 | 41.60 | 44.00 | 15.00 |
> | LLaVA-1.5-13B + ETA | 15.09 | 11.67 | 22.60 | 20.80 | 12.50 |
> | Llama3.2-11B-Vision-Instruct + ETA | **2.64** | **3.99** | **8.20** | **3.20** | **7.50** |

---

> ### Author Response · Authors · 2024-11-21
> **Response to Reviewer tnBM (3/3)**
>
> > - Weakness 3: Pre-generation Evaluator: although the image contains the harmful image, the textual instruction might be normal and benign (e.g., a criminal scene image + "How to prevent such crimes?"). According to the proposed method, it may judge these benign inputs as unsafe, which possibly leaves as a side-effect.
>
> Thanks for the valuable question. In the scenario you described, **ETA does not interfere with the model’s normal responses**. It is indeed likely that the Pre-Generation Evaluator classifies the input as unsafe. However, if the text instruction is beneficial, such as “How to prevent such crimes?”, the **Post-Generation Evaluator can classify the response as “safe” if it provides a safe and helpful answer**. Our criteria are designed such that the alignment phase is only triggered when both the Pre- and Post-Generation Evaluators classify the response as unsafe; otherwise, the model’s original reply is directly output. As analyzed in Section 4.1.2 and demonstrated in Table 10, this setup allows for accurate judgments to handle such cases effectively.
>
> To further verify whether ETA might have potential side effects under the reviewer’s setting, we used the image in Fig. 1, which **depicts “Defensive Driving”** classified as an **unsafe image** by ETA’s Pre-Generation Evaluator. We input this image along with the prompt “How to prevent such crimes?”. ETA **directly outputs the response generated by the baseline**, and we report the response below.
>
>       To prevent crimes like the one depicted in the image, where a car is being used to commit a crime, it is essential to implement various preventive measures. These can include:
>
>      1. Strengthening law enforcement: Increasing the presence of police officers and patrolling areas with high crime rates can deter potential criminals.
>      ...
>
> > - Weakness 4: Sentence-level Deep Alignment: since the proposed deep alignment selects the optimal sentence under the consideration of the safeness of the given sentence and its correlation with the input image in a greedy manner, the resultant sequence of sentences might not be ensured to be logically connected. This potential side-effect might rather hinder the reliability of the VLMs.
>
> Thank you for your valuable questions. ETA **does not cause semantic or logical incoherence**. Our method **does not** generate k full responses and then select the best candidate sentence for each individual sentence. This approach could indeed affect the logical coherence of the generated response. Instead, we start by generating k candidate sentences. For the first sentence, we select the highest-scoring sentence out of k candidates. Then, **following this chosen sentence**, we generate k candidates as the second sentence and select the one with the highest score. At this point, the **response consists of two sentences**. The process continues in this manner, guiding the generation until it is complete.
>
> It is important to note that in the sentence-level Best-of-N approach, **each candidate in the i-th generation** is generated **based on the first i-1 sentences**. This ensures that the context established by the previous sentences is taken into account when generating the next one, preserving the coherence and logical flow of the response. Some papers have verified that methods similar to beam search do not affect semantic and logical coherence, such as [2]. We have updated the manuscript with a more detailed explanation.
>
> ## Reference
> [1] Alec Radford, Jong Wook Kim, Chris Hallacy, et al. Learning transferable visual models from natural language supervision. In International conference on machine learning, pp. 8748–8763. PMLR, 2021.
>
> [2] Zhanhui Zhou, Zhixuan Liu, Jie Liu, et al, Weak-to-Strong Search: Align Large Language Models via Searching over Small Language Models. NeurIPS 2024

---

> ### Author Response · Authors · 2024-11-25
> **A kindly reminder**
>
> ​Dear Reviewer,
>
> Thank you again for your valuable feedback and questions. We have provided responses addressing your concerns and uploaded a new version of our manuscript.
>
> As the discussion period draws to a close, we kindly encourage you to **take a look at our responses** and **let us know if you have any remaining questions or concerns**.
>
> We sincerely appreciate your time and effort in helping us improve our work.
>
> Sincerely,
>
> Authors

---

> ### Comment · Reviewer_tnBM · 2024-11-26
>
> Thanks for the detailed responses. Almost all of my concerns were resolved, except for the novelty issue (simple leverage of CLIP and reward models) as the other reviewers mentioned. I raised my score to 5.

---

### Official Review · Reviewer_8ihd · 2024-11-02

**Soundness:** 2
**Presentation:** 3
**Contribution:** 2
**Rating:** 5
**Confidence:** 3

**Summary:**

The paper explores solutions for enhancing the safety of Vision Language Models (VLMs). The authors introduce a strategy called Evaluating Then Aligning (ETA) to address this issue at inference time.

ETA first detects unsafe visual inputs using CLIP similarity to an unsafety-specific prompt (pre-generation score) and an LLM-based reward model to assess initial output safety (post-generation score). If both scores indicate that the input is unsafe (pre-generation score is above the threshold, post-generation score is below the threshold), ETA appends a predefined prefix (e.g., "As an AI assistant,") to the prompt, guiding the VLM to generate a safer response by maximizing the weighted sum of two factors: the similarity between the current output sentence and the visual input, and the post-generation score of the previously generated sentence.

The authors conduct experiments across various LLMs and datasets to demonstrate the effectiveness of the proposed strategy.

**Strengths:**

1.	This paper introduces a straightforward, training-free strategy to address safety issues in VLMs.
2.	Experiments conducted across multiple models and datasets demonstrate the strategy's effectiveness and generalizability.
3.	The paper is well-structured, providing a clear narrative that effectively outlines both the problem and solution.

**Weaknesses:**

1.	The setting of the helpfulness experiment (Table 2) is not well rigorous.
The experiment is conducted using general comprehensive benchmarks and VQA datasets, which is not fully consistent with the "usefulness" goal the authors aim to address—namely, ensuring helpful responses even with unsafe inputs. While Table 3 provides some helpfulness results, the range of benchmarks used is somewhat limited. Testing on more models and datasets can better verify the effectiveness of this strategy.
2.	The analysis of the ablation study results is inadequate:
When the Utility Guide component is removed from the strategy, the unsafe rate actually decreases, yet this result is not fully analyzed. Does this suggest that the Utility Guide might be unnecessary? A deeper exploration is needed to clarify its role and significance.
3.	The proof of the Motivation in the MULTIMODAL EVALUATION (i.e. Continuous Visual Token Embeddings Bypass Safety Mechanisms) is not very rigorous: In the proof experiment, continuous visual embeddings were converted to the nearest discrete text embeddings, showing a 7% reduction in unsafe rates on one dataset. Testing this approach across more datasets and models could strengthen its reliability.
4.	The effectiveness of the strategy is not well proven. The number of baselines (especially for inference-time baselines) is too small to prove the effectiveness of the strategy.

**Questions:**

Please address the concerns in the Weakness part.

---

> ### Author Response · Authors · 2024-11-13
> **Mismatched Reviewer Comments**
>
> Dear PCs, SACs, ACs, and Reviewer 8ihd,
>
> We have noticed that the comments provided by Reviewer 8ihd appear to correspond to a different paper, as we did not propose DynVLA in submission 675. We would greatly appreciate it if you could kindly review this issue and have the reviewer resubmit the correct comments for our submission.
>
> Thank you very much for your time and assistance.
>
> Best,
>
> Authors of submission 675

---

> > ### Comment · Reviewer_8ihd · 2024-11-13
> >
> > Hi,
> >
> > I mixed up the comments for two articles while copying them from a Word document. I am sorry about this confusion, and I have corrected the review.

---

> ### Author Response · Authors · 2024-11-21
> **Response to Reviewer 8ihd (1/2)**
>
> We would like to thank the reviewer for the thoughtful and thorough comments on our paper as well as for recognizing our presentation, the effective and generalizable framework of our ETA, and the detailed and rich experiments. We answer your questions below.
>
> > - Weakness 1: The setting of the helpfulness experiment (Table 2) is not well rigorous. The experiment is conducted using general comprehensive benchmarks and VQA datasets, which is not fully consistent with the "usefulness" goal the authors aim to address—namely, ensuring helpful responses even with unsafe inputs. While Table 3 provides some helpfulness results, the range of benchmarks used is somewhat limited. Testing on more models and datasets can better verify the effectiveness of this strategy.
>
> Thanks for the constructive comment. We evaluate helpfulness/usefulness from two aspects in this paper.
>
> (i) **Helpfulness/Usefulness under Unsafe Settings:** To evaluate the helpfulness in unsafe scenarios, we followed the setup from [1], which is the **only benchmark assessing helpfulness in unsafe settings**. The results, shown in Table 3 and 4, demonstrate that ETA significantly enhances both the safety and helpfulness of models in such settings. Please note that the evaluation metrics in Table 3 and 4 are based on the **GPT-4 API**, which is **extremely costly**. Therefore, we did not conduct experiments on more models. We believe that the **96.6% Helpfulness Win-Tie rate** compared to the baseline on LLaVA-1.5-7B is already strong evidence to demonstrate the superiority of our ETA. We have also provided examples of helpful responses by ETA under unsafe settings in the Appendix. For instance, as illustrated in the case study in Figure 10, ETA can offer helpful suggestions like: “Instead, I can share general advice to protect cars and their contents from potential thefts: ……”
>
> (ii) **Helpfulness/Usefulness under General Settings:** We observed that existing methods, including fine-tuning-based and inference-time-based approaches, tend to **degrade the general capabilities of models**, as reflected in the comprehensive benchmarks and VQA datasets you mentioned. We provide the results of applying ETA to VLM baselines in Table 2 and Table 8, along with a comprehensive analysis in Section 5.2 that highlights the helpfulness of ETA in this setting.
>
> > - Weakness 2: The analysis of the ablation study results is inadequate: When the Utility Guide component is removed from the strategy, the unsafe rate actually decreases, yet this result is not fully analyzed. Does this suggest that the Utility Guide might be unnecessary? A deeper exploration is needed to clarify its role and significance.
>
> Thank you for the insightful comment.
>
> (i) It is worth noting that **the scales and evaluation methods of USR and the Helpful Score in Table 5 are different**. USR is calculated by $\text{USR}=\frac{|\{\text{unsafe responses}\}|}{|\{\text{all responses}\}|}$, while the Helpful Score represents the average score given by GPT-4 for all responses, with a maximum score of **10**. If we normalize the Helpful Score to match the scale of USR, the improvement in Helpful Score from the fourth row to the fifth row, with the addition of the Utility Guide, corresponds to an increase of 1.20. However, the increase in USR is only 0.38. This suggests that the Utility Guide significantly enhances the helpfulness of responses but has only a slight impact on their safety capabilities.
>
> (ii) We used the UnSafe Rate (USR) evaluated using **MD-Judge-v0.2-internlm2_7B in our paper**. To further verify the effect of the Utility Guide, we also report the Attack Success Rate **(ASR) assessed through key-string matching** and the **USR evaluated by Llama-Guard-3-8B** in the table below, which are another two common metrics for evaluating the harmfulness of responses.
>
>
> | Method  | USR (MD-Judge-v0.2-internlm2_7B) | USR (Llama-Guard-3-8B) | ASR (Key-String Match) | Helpful Score |
> | :------- | :----: | :----: | :----: | :----: |
> | Shallow Alignment + Safety Guide Deep Alignment | **16.60** | 7.17 | 38.87 | 8.38 |
> | Shallow Alignment + Safety Guide & Utility Guide Deep Alignment | 16.98 | **6.80** | **38.87** | **8.50** |
>
> It can be observed that different evaluation methods yield consistent conclusions: **Integrating the utility score defined in Eq. 5 into deep alignment can significantly enhance the helpfulness of responses without compromising the model’s safety capabilities.** This shows the importance of utility score in deep alignment. We have added the above explanation about the ablation study in our manuscript.

---

> ### Author Response · Authors · 2024-11-21
> **Response to Reviewer 8ihd (2/2)**
>
> > - Weakness 3: The proof of the Motivation in the MULTIMODAL EVALUATION (i.e. Continuous Visual Token Embeddings Bypass Safety Mechanisms) is not very rigorous: In the proof experiment, continuous visual embeddings were converted to the nearest discrete text embeddings, showing a 7% reduction in unsafe rates on one dataset. Testing this approach across more datasets and models could strengthen its reliability.
>
> Thank you for the constructive suggestion. To further support our conclusion, we have additionally evaluated this approach on the SPA-VL Harm test set and VLSafe. The VLSafe dataset contains a total of 1,100 data samples. Due to the significant time required for the experiments, we randomly sampled 100 data points for testing. In the finalized version, we plan to include the complete results and we believe they will be similar. Additionally, we tested four baseline models on these two datasets, with the results presented in the table below. The **decrease in USR** after applying the mapping supports our motivation: "**Continuous visual token embeddings bypass safety mechanisms (which are aligned on discrete text token embeddings)**."
>
>
>
> | Method  | SPA-VL Harm | VLSafe |
> | :------- | :----: | :----: |
> | LLaVA-1.5-7B | 46.04 | 78.00 |
> | + Continuous to Discrete | **39.25** | **40.00** |
> | LLaVA-1.5-13B | 40.75 | 61.00 |
> | + Continuous to Discrete | **24.91** | **41.00** |
> | InternVL-Chat-1.0-7B | 46.79 | 77.00 |
> | + Continuous to Discrete | **35.09** | **47.00** |
> | InternLM-XComposer-2.5-7B | 27.55 | 15.00 |
> | + Continuous to Discrete | **21.51** | **7.00** |
>
> > - Weakness 4: The effectiveness of the strategy is not well proven. The number of baselines (especially for inference-time baselines) is too small to prove the effectiveness of the strategy.
>
> Thank you for the valuable comment. Considering that inference-time alignment strategies are still underexplored in the field of VLMs, ECSO is the **only** open-sourced method that **does not require additional data or training, making it a plug-and-play solution similar to ETA**. Some methods are categorized as inference-time approaches but require **additional data and training phases**, like [2] and [3]. Therefore, we believe ECSO is the most comparable method to ETA. To further demonstrate the effectiveness of ETA, we apply it to three more recent and advanced VLM baselines: LLaVA-NeXT-8B, LLaVA-OneVision-7B-Chat, and Llama3.2-11B-Vision-Instruct. The results, shown in the table below, indicate that even for Llama3.2-Vision-11B-Instruct, which has already been aligned via RLHF, our ETA successfully decreases the USR across five different test settings. This highlights the superiority of ETA in enhancing VLM safety.
>
> | Method  | SPA-VL Harm | MM-SafetyBench | FigStep (Vanilla) | FigStep (Suffix) | Adv. Image+Text |
> | :------- | :----: | :----: | :----: | :----: | :----: |
> | LLaVA-NeXT-8B | 23.02 | 30.18 | 49.40 | 63.40 | 62.50 |
> | + ETA | **11.32** | **10.48** | **20.60** | **19.60** | **17.50** |
> | LLaVA-OneVision-7B-Chat | 15.85 | 29.76 | 45.20 | 40.40 | 70.00 |
> | + ETA | **6.79** | **10.60** | **16.80** | **19.40** | **20.00** |
> | Llama3.2-11B-Vision-Instruct | 7.17 | 19.17 | 41.60 | 44.00 | 15.00 |
> | + ETA | **2.64** | **3.99** | **8.20** | **3.20** | **7.50** |
>
> In addition, we have summarized the extensive experiments conducted in our study, which include four baseline models, four safety-related datasets, and seven helpfulness-related datasets, all of which demonstrate the remarkable effectiveness of ETA:
>
> (i) **ETA Enhances the Safety Capability of VLMs:** In Table 1, 6, and 7, we demonstrate that ETA not only significantly decreases the USR across all four baselines but also outperforms ECSO.
>
> (ii) **ETA Does Not Compromise the General Ability of VLMs:** In Table 2, 3, and 8, we show that compared to ECSO, our ETA maintains the models' general abilities and provides more helpful suggestions in unsafe settings.
>
> (iii) **Both the Evaluation and Alignment Phases of ETA Outperform the ECSO:** In Table 9, to further highlight the effectiveness of both the evaluating and aligning phases of our ETA, we replace our **multimodal evaluator** with the **self-evaluation approach used in ECSO**. Comparing "ECSO Eval. & ETA Align." with "ETA" we observe that the complete ETA process not only provides safer responses but also maintains the models' general capabilities.
>
> ## Reference
> [1] Yongting Zhang, Lu Chen, Guodong Zheng, and et al. Spa-vl: A comprehensive safety preference alignment dataset for vision language model. arXiv preprint arXiv:2406.12030, 2024c.
>
> [2] Renjie Pi, Tianyang Han, Yueqi Xie, et al. Mllm-protector: Ensuring mllm’s safety without hurting performance. arXiv preprint arXiv:2401.02906, 2024.
>
> [3] Yunhao Gou, Kai Chen, Zhili Liu, et al. Eyes closed, safety on: Protecting multimodal llms via image-to-text transformation. arXiv preprint arXiv:2403.09572, 2024.

---

> ### Author Response · Authors · 2024-11-25
> **A kindly reminder**
>
> ​Dear Reviewer,
>
> Thank you again for your valuable feedback and questions. We have provided responses addressing your concerns and uploaded a new version of our manuscript.
>
> As the discussion period draws to a close, we kindly encourage you to **take a look at our responses** and **let us know if you have any remaining questions or concerns**.
>
> We sincerely appreciate your time and effort in helping us improve our work.
>
> Sincerely,
>
> Authors

---

> ### Author Response · Authors · 2024-12-01
> **A kindly reminder**
>
> Dear Reviewer,
>
> Thank you once again for your insightful feedback and questions. We have provided detailed explanations and additional experimental results addressing your concerns and uploaded a new version of our manuscript. The contents included in the response are:
>
> - **(i)** Explanation about the **setting of helpfulness/usefulness** in our experiments
>
> - **(ii)** Experiments and explanation of the **utility guide’s ablation study**
>
> - **(iii)** Experiments on **additional baselines and datasets to support the motivation**
>
> - **(iv)** Experiments on **more VLM baselines to demonstrate the effectiveness of ETA**
>
>
> As the discussion period is about to end, we kindly remind you to **review our response** to see if it addresses your concerns. **If you still have any questions or concerns, please let us know**.
>
> We sincerely appreciate your time and effort in helping us improve our work.
>
> Sincerely,
>
> Authors

---

### Official Review · Reviewer_Y2F1 · 2024-11-03

**Soundness:** 3
**Presentation:** 3
**Contribution:** 3
**Rating:** 8
**Confidence:** 4

**Summary:**

The paper introduces ETA, a two-phase framework for enhancing Vision Language Model (VLM) safety at inference time. The proposed method assesses visual and textual safety using CLIP scores for images and a reward model for text. It also applies a "safety prefix" (shallow alignment) and best-of-N search (deep alignment) to ensure responses are harmless and relevant. The ETA significantly reduced unsafe responses through comprehensive experiments.

**Strengths:**

1. The idea of the method is intuitive and easy to follow.
2. The method is plug-and-play, could be deployed to many VLM systems.
3. Comprehensive experiments on some VLMs with good performance.

**Weaknesses:**

1. The method itself, is more like a combinaiton of pre-processing and post-processing techniques. The contribution of this method maybe limited.
2. As a plug-and-play method, I expect to see it to be applied to more recent VLMs, such as LLaVA-Next, LLaMA3.2.
3. Also I would expect to see the model performance drop on more recent and widely-used benchmarks like MMMU
4. Some highly-relevant references about VLM/LLM safety are missing [1][2]

I would be happy to raise my score once my concerns are addressed.

[1] Inan H, Upasani K, Chi J, et al. Llama guard: Llm-based input-output safeguard for human-ai conversations[J]. arXiv preprint arXiv:2312.06674, 2023.

[2] Tu H, Cui C, Wang Z, et al. How many unicorns are in this image? a safety evaluation benchmark for vision llms[C]. ECCV, 2024.

**Questions:**

Is there any compromised approach to also employ similar strategy to API-based VLM/LLMs like GPT-4(V)?

---

> ### Author Response · Authors · 2024-11-21
> **Response to Reviewer Y2F1 (1/3)**
>
> We sincerely thank the reviewer for their thoughtful and thorough comments on our paper. We deeply appreciate the recognition of ETA's intuitive design, plug-and-play versatility, and comprehensive experiments demonstrating strong performance. We answer your questions below.
>
> > - Weakness 1: The method itself, is more like a combination of pre-processing and post-processing techniques. The contribution of this method maybe limited.
>
> While ETA involves pre-processing and post-processing, its contribution lies in how these elements are uniquely combined to address VLM safety. In fact, ensuring safety while preserving the original capabilities of VLMs is a very challenging problem, and we believe this is where ETA makes a significant contribution. We consider the simplicity of ETA an advantage rather than a limitation. We summarize the contributions of ETA as follows:
>
> (i) **New Perspective on Safety Issues in VLMs:** We offer a new analysis on the failure of existing defense mechanisms in VLMs, and show that the primary factor contributing to safety problems in VLM systems is the continuous nature of visual token embeddings.
>
> (ii) **First to Introduce Additional Visual Safeguards:** The usage of CLIP in our work differs significantly from its application in prior studies. Previous works have primarily leveraged the CLIP score during post-processing stages, such as inference and evaluation, to assess the similarity between output text and input images. In contrast, our paper innovatively utilizes the CLIP model during the pre-processing stage. We demonstrate that the CLIP score can be effectively generalized to evaluate the **safety** of continuous visual embeddings, which can easily bypass the existing safety mechanisms in VLMs. By utilizing predefined prompts with unsafe semantics, we show that it is generalizable to classify input images for safety directly, providing a novel application of CLIP in ensuring the safety of visual inputs.
>
> (iii) **An innovative Strategy to Improve Safety Evaluation Capability of Reward Models:** We demonstrate that designing a safety-specific input format for general reward models can significantly enhance their ability to more effectively evaluate the safety of responses. This differs from RMs’ usage in prior studies.
>
> (iv) **Harmless and Helpful Responses, Rather Than Generic Refusals:** We found that existing methods often provide generic refusals such as “I cannot help you, …”. However, an ideal response should be both harmless and helpful, offering helpful suggestions to users. By leveraging the multimodal evaluator (CLIP+RM) for sentence-level Best-of-N searching, we can effectively identify safer and more helpful generation trajectories. This is also a novel usage of CLIP and RM.
>
> (v) **Superior Performance of ETA as a Simple Plug-and-Play Framework:** Extensive experiments demonstrate that our plug-and-play approach is applicable to common VLM baselines, significantly enhancing the model’s safety without affecting its general capabilities. Moreover, it improves both the safety and usefulness of the model’s responses in unsafe scenarios.

---

> ### Author Response · Authors · 2024-11-21
> **Response to Reviewer Y2F1 (2/3)**
>
> > - Weakness 2: As a plug-and-play method, I expect to see it to be applied to more recent VLMs, such as LLaVA-NeXT, Llama3.2.
>
> Considering that LLaVA-NeXT-8B (**released in May 2024**) and Llama3.2-11B-Vision-Instruct (**released in Sep. 2024**) are very recent models, they were not included as baselines in the paper. However, we did include another recent and strong VLM, InternLM-XComposer-2.5-7B (**released in July 2024**) in the paper. **To further verify the applicability of our method**, we have now tested the effectiveness of ETA on the suggested recent models **LLaVA-NeXT-8B** (**released in May 2024**), **LLaVA-OneVision-7B-Chat** (**released in Sep. 2024**), and **Llama3.2-11B-Vision-Instruct** (**released in Sep. 2024**).
>
> | Method  | SPA-VL Harm | MM-SafetyBench | FigStep (Vanilla) | FigStep (Suffix) | Adv. Image+Text |
> | :------- | :----: | :----: | :----: | :----: | :----: |
> | LLaVA-NeXT-8B | 23.02 | 30.18 | 49.40 | 63.40 | 62.50 |
> | + ETA | **11.32** | **10.48** | **20.60** | **19.60** | **17.50** |
> | LLaVA-OneVision-7B-Chat | 15.85 | 29.76 | 45.20 | 40.40 | 70.00 |
> | + ETA | **6.79** | **10.60** | **16.80** | **19.40** | **20.00** |
> | Llama3.2-11B-Vision-Instruct | 7.17 | 19.17 | 41.60 | 44.00 | 15.00 |
> | + ETA | **2.64** | **3.99** | **8.20** | **3.20** | **7.50** |
>
> The results of Llama3.2-11B-Vision-Instruct demonstrate that, despite the baseline being aligned through RLHF and equipped with a safety-aware module (Llama Guard), ETA still enhances the baseline's safety capabilities, particularly under more severe attack settings (e.g., suffix attacks). Moreover, a comparison of the results for LLaVA-OneVision-Chat-7B and Llama3.2-11B-Vision-Instruct on FigStep reveals that models with stronger instruction-following capabilities and RLHF alignment are more likely to produce safer responses during ETA's alignment phase. **This highlights the promising potential of ETA for application in the future, more robust, and safer VLMs.**
>
> > - Weakness 3: Also I would expect to see the model performance drop on more recent and widely-used benchmarks like MMMU.
>
> Thank you for the constructive suggestion. In our experiments, we evaluate several recent comprehensive benchmarks, such as MME and widely-used VQA benchmarks, to ensure that our ETA does not negatively impact the baseline models’ general performance. To assess whether ETA weakens baseline performance on more recent and complex benchmarks, **we test two different settings of MMMU-Pro (robust version of MMMU) (standard 4 options, and vision) using direct prompts.** The results in the table below demonstrate that, even on more complex and recent benchmarks, ETA still does not affect the baseline models' overall helpfulness.
>
> | Method  | Standard 4 options | Vision |
> | :------- | :----: | :----: |
> | LLaVA-1.5-7B | 35.38 | 12.66 |
> | + ETA | 35.38 | 12.66 |
> | LLaVA-1.5-13B | 33.18 | 12.49 |
> | + ETA | 33.18 | 12.49 |
> | InternVL-Chat-1.0-7B | 33.01 | 11.62 |
> | + ETA | 33.01 | 11.62 |
> | LLaVA-NeXT-8B | 35.61 | 12.43 |
> | + ETA | 35.61 | 12.43 |
> LLaVA-OneVision-7B-Chat | 43.06 | 15.61 |
> | + ETA | 43.06 | 15.61 |
> | Llama3.2-11B-Vision-Instruct | 43.76 | 15.66 |
> | + ETA | 43.76 | 15.66 |
>
> > - Weakness 4: Some highly-relevant references about VLM/LLM safety are missing [1][2]
>
> Thanks for the detailed comments. We have included these references in the related work section in the revision.

---

> ### Author Response · Authors · 2024-11-21
> **Response to Reviewer Y2F1 (3/3)**
>
> > - Questions: Is there any compromised approach to also employ similar strategy to API-based VLM/LLMs like GPT-4(V)?
>
> Thanks for the valuable question. We did not include experiments on API-based models, as the baselines we compared did not consider the use case of API-based models. Theoretically, **the ETA framework is applicable to API-based models.** However, considering the **extremely high costs** of API calls, we leave further investigation for future work. Here, we briefly explain how ETA can be applied to API-based VLMs/LLMs. First, the evaluation phase can be applied to API-based VLMs/LLMs, as this phase only requires inputting the image and the standard output from the API model to complete the process. Second, for the behavior classified as safe, we can directly output the response. For behaviors classified as unsafe, some API-based VLMs/LLMs, such as GPT-3.5-Turbo-Instruct, which are **compatible with the legacy completions endpoint and not Chat Completions**, can integrate ETA by appending a safety prefix to the end of the input. This setup enables the generation of sentence candidates, which can then be filtered using our deep alignment. We hypothesize that the complete process will be similar to the chunk-level beam search approach used in experiments on GPT-3.5-Turbo-Instruct as described in [6].
>
> ## Reference
>
> [1] Inan H, Upasani K, Chi J, et al. Llama guard: Llm-based input-output safeguard for human-ai conversations[J]. arXiv preprint arXiv:2312.06674, 2023.
>
> [2] Tu H, Cui C, Wang Z, et al. How many unicorns are in this image? a safety evaluation benchmark for vision llms[C]. ECCV, 2024.
>
> [3] Hannah Brown, Leon Lin, Kenji Kawaguchi, and Michael Shieh. Self-evaluation as a defense against adversarial attacks on llms. arXiv preprint arXiv:2407.03234, 2024b.
>
> [4] Yunhao Gou, Kai Chen, Zhili Liu, et al. Eyes closed, safety on: Protecting multimodal llms via image-to-text transformation. arXiv preprint arXiv:2403.09572, 2024.
>
> [5] Renjie Pi, Tianyang Han, Yueqi Xie, et al. Mllm-protector: Ensuring mllm’s safety without hurting performance. arXiv preprint arXiv:2401.02906, 2024.
>
> [6] Zhanhui Zhou, Zhixuan Liu, Jie Liu, et al, Weak-to-Strong Search: Align Large Language Models via Searching over Small Language Models. NeurIPS 2024

---

> > ### Comment · Reviewer_Y2F1 · 2024-11-21
> >
> > Thank you for your response and additional results. My concerns are addressed. I've decided to raise the score.

---

> > > ### Author Response · Authors · 2024-11-22
> > >
> > > Thanks a lot for your positive feedback. Your insightful comments have greatly improved our work. We sincerely appreciate your support.

---

### Author Response · Authors · 2024-11-21

Dear PCs, SACs, ACs, and Reviewers,

We would like to thank you for your valuable feedback and insightful reviews, which have greatly contributed to improving the paper. This is **a clear and well-written** (Reviewer 8ihd, Reviewer tnBM, Reviewer JJRX) manuscript with an **intuitive and novel idea** (Reviewer Y2F1, Reviewer JJRX), we proposed an **effective and superior plug-and-play framework** (Reviewer Y2F1, Reviewer 8ihd, Reviewer tnBM, Reviewer JJRX), the **systematically, and Comprehensive** experiments on **multiple baselines and datasets** validate **ETA’s effectiveness and superiority** (Reviewer Y2F1, Reviewer JJRX).

In our rebuttal, we addressed the following raised concerns/misunderstandings.

- We have clarified the contribution, novelty, and motivation of our method.
- We have validated ETA's performance on three more recent and strong VLM baselines: **LLaVA-NeXT-8B** (released in May 2024), **LLaVA-OneVision-7B-Chat** (released in Sep. 2024), and **Llama3.2-11B-Vision-Instruct** (released in Sep. 2024).
- We have provided the performance of ETA on a more recent and complex comprehensive benchmark **MMMU-Pro**.
- We have conducted the experiment of **continuous visual token embeddings to discrete text token embeddings** on more baselines and datasets to clarify our motivation.
- We have provided a more detailed explanation of the role of the **utility guide** in the ablation study and illustrations of why ETA **does not** have some potential side effects.

We would like to highlight the novelty and contributions of our work. Ensuring safety while preserving the original capabilities of VLMs is a highly challenging problem, and this work presents a plug-and-play solution to address it.
- We identify that the key issue of VLM safety lies in the continuous nature of visual token embeddings.
- Noting that continuous visual embeddings in VLMs are essentially obtained by the CLIP encoder, we innovatively use CLIP to assess the safety of visual inputs. This differs from the existing usage of CLIP in VLMs, which typically focuses on measuring the similarity between text output and visual inputs.
- We develop a bi-level alignment strategy for addressing detected unsafe behaviors, which uses CLIP and the reward model as guidance to steer the text output to be both safe and useful.
- Extensive experiments demonstrate that our approach applies to common VLM baselines, significantly enhancing the model’s safety without affecting its general capabilities.

We believe these insights, combined with a simple and effective approach and significant empirical improvement, provide a valuable contribution to the field of VLM safety. We hope that our responses address your questions and concerns. If there are any additional insights, questions, or clarifications on our responses/submission that you would like to discuss with us, we would be very grateful to hear them, your feedback is invaluable for the improvement of our research.

Best,

Authors of Submission 675

---

### Author Response · Authors · 2024-11-22

Dear PCs, SACs, ACs, and Reviewers,

We have updated the PDF to reflect the corresponding revisions (highlighted in gray blue).

Thank you!

Best,

Authors of Submission 675

---

### Meta-Review · Area_Chair_x4Md · 2024-12-23

**Metareview:**

This paper investigates the safety aspect of VLMs. The key contribution lies in the development of a two-phase inference-time alignment framework, where the first stage assesses the safety of input visual content and initial output responses, and the second stage involves a multi-level alignment strategy. The provided emprical results well support the effectiveness of the proposed method.

Overall, the reviewers found this paper well-written and easy to follow, and they appreciated both the simplicity and effectiveness of the proposed method. Nonetheless, some major concerns are also raised, including: 1) the novelty is somewhat limited; 2) the current motivation is not strongly supported, and more baselines and datasets should be included; 3) the main experiment should consider more recent VLMs; and 4) some experimental results require further analysis and discussion.

During the rebuttal and discussion stages, the authors provided detailed responses, additional experiments, and clarifications. Two reviewers acknowledged most of their concerns are well addressed, and agreed on accepting it. Reviewer tnBM acknowledged that the only remaining concern is the novelty (a point also raised by Reviewer JJRX). After reviewing the authors' rebuttal and clarification on novelty, the AC believes its contribution is sufficiently different from existing works and found its novelty acceptable (though not substantial). Reviewer 8ihd did not engage in the discussion, but after reviewing their review and the authors' responses, the AC is confident that their concerns were adequately addressed.

Given the importance of this research topic, the strong performance of this plug-and-play solution, and the fact that most concerns (except for the "minor" novelty issue) have been satisfactorily addressed, the AC recommends accepting the submission.

**Additional Comments On Reviewer Discussion:**

The major concerns are summarized in the meta-review.  Overall, the rebuttal effectively addresses points (2)–(4). Regarding point (1) (i.e., novelty), the AC considers it a minor issue, which is outweighed by the paper's other merits like strong empirical performance. Therefore, the final decision is acceptance.

---

### Decision · Program_Chairs · 2025-01-22

Accept (Poster)